

# The effect of differential spatiotopic information on the acquisition and generalization of fear of movement-related pain

Ann Meulders[1,2] and Johan W. Vlaeyen[1,2]

[1] Experimental Health Psychology, Maastricht University, Maastricht, Netherlands
[2] Research Group Health Psychology, Katholieke Universiteit Leuven, Leuven, Belgium

Corresponding author
Ann Meulders,
ann.meulders@kuleuven.be

## ABSTRACT

Fear of movement-related pain significantly contributes to musculoskeletal chronic pain disability. Previous research has shown that fear of movement-related pain can be classically conditioned. That is, in a differential fear conditioning paradigm, after (repeatedly) pairing a neutral joystick movement (conditioned stimulus; CS+) with a painful stimulus (unconditioned stimulus; pain-US), that movement in itself starts to elicit self-reported fear and elevated psychophysiological arousal compared to a control joystick movement (CS−) that was never paired with pain. Further, it has been demonstrated that novel movements that are more similar to the original CS+ elicit more fear than novel movements that are more similar to the CS−, an adaptive process referred to as *stimulus generalization*. By default, movement/action takes place in reference to the three-dimensional space: a movement thus not only involves proprioceptive information, but it also contains spatiotopic information. Therefore, the aim of this study was to investigate to what extent spatiotopic information (i.e., endpoint location of movement) contributes to the acquisition and generalization of such fear of movement-related pain besides proprioception (i.e., movement direction). In a between-subjects design, the location group performed joystick movements from the middle position to left and right; the movement group moved the joystick from left and right to the middle. One movement (CS+) was paired with pain, another not (CS−). *Feature overlap* between CSs typically reduces differential learning. The endpoint of both CSs in the movement group is an overlapping feature whereas in the location group the endpoint of both CSs is distinct; therefore we hypothesized that there would be less differential fear learning in the movement group compared to the location group. We also tested generalization to movements with similar proprioceptive features but different endpoint location. Following the principle of stimulus generalization, we expected that novel movements in the same direction as the CS+ but with a different endpoint would elicit more fear than novel movement in the same direction of the CS− but with a different endpoint. Main outcome variables were self-reported fear and pain-US expectancy and eyeblink startle responses (electromyographic). Corroborating the feature overlap hypothesis, the location group showed greater differential fear acquisition. Fear generalization emerged for both groups in the verbal ratings, suggesting that fear indeed accrued to proprioceptive CS features; these effects, however, were not replicated in the startle measures.

## INTRODUCTION

Accumulating empirical evidence, mostly from studies with chronic musculoskeletal pain patients, shows that *pain-related fear* significantly predicts physical performance, functional disability (*Swinkels-Meewisse et al., 2006*), and sick leave (*Gheldof et al., 2005*) and that such fear is often more disabling than the pain itself (*Crombez et al., 1999*). Previous research has convincingly demonstrated the role of associative learning processes in the development (*Meulders, Vansteenwegen & Vlaeyen, 2011*, *2012*) and spreading (*Geschwind et al., 2015*; *Meulders, Jans & Vlaeyen, 2015*; *Meulders et al., 2013*; *Meulders & Vlaeyen, 2013*) of fear of movement-related pain. More particularly, an initially neutral proprioceptive stimulus associated with a particular movement (conditioned stimulus, CS; e.g., bending forward left to pick something from the floor) that is paired with pain (unconditioned stimulus, pain-US; e.g., a shooting pain in the lower back) may start to elicit protective responding in the anticipation of pain such as heightened attentional orientation, autonomic arousal, and avoidance behavior compared with another neutral movement (CS−; bending forward right) that was not paired with pain. Yet, clinical observations clearly show that chronic pain patients do not only avoid those movements that were paired with pain in an initial (acute pain) learning episode, but that fear of movement-related pain and its associated avoidance behavior can generalize to other movements that were never paired with pain. This learning mechanism, referred to as *stimulus generalization*, is adaptive because it enables us to extrapolate information from one aversive learning episode and apply it to novel, similar situations without having to learn everything anew (*De Clercq et al., 2006*; *Ghirlanda & Enquist, 2003*; *Honig & Urcuioli 1981*; *Kalish, 1969*). However, when these conditioned protective responses spread to safe stimuli, fear and avoidance lose their adaptive function (*Meulders et al., 2014*, *2017*; *Meulders, Jans & Vlaeyen, 2015*).

The ability to predict *which movements/actions* are accompanied or followed by pain has an evolutionary advantage, as it enables us to initiate appropriate protective action (*Vlaeyen, 2015*). Additionally, the ability to precisely localize sensory stimuli –providing information about *where* pain will occur– is also fundamental to how, we interact with our environment; indeed, to our survival. Spatiotopic information is thus equally important for threat appraisal and defensive action (*De Paepe, Crombez & Legrain, 2015*). In real-life situations, spatiotopic, and movement-related (proprioceptive) information always interact with each other; all movement/action takes place in reference to the three-dimensional space. With respect to the prediction of the pain both channels of information may diverge or converge. In the above-mentioned bending example, both bending forward left (CS+) and bending forward right (CS−) entails moving the head facedown, in terms of planes of reference that is, moving toward the inferior part of the cross-sectional plane. Although the end point location of both movements is quite similar, the proprioceptive characteristics of these movements are dissimilar. In this example

the CS+ and CS− thus share the same endpoint location in the three-dimensional space (facedown). In learning theory this can be conceptualized as feature overlap. *Feature overlap* between the CS+ and CS− has been shown to reduce differential learning. For example, *Haddad et al. (2012)* showed that conditioned fear generalized more to a CS− (female face) that resembled the CS+ (another female face), compared to a perceptually dissimilar CS− (gray oval of similar proportions to the female faces).

In this study, we wanted to address the intriguing question as to what extent spatiotopic information contributes to the acquisition of fear of movement-related pain in addition to proprioceptive or movement-related information only. We used an adapted version of the Voluntary Joystick Movement Paradigm (VJMP) (*Meulders, Vansteenwegen & Vlaeyen, 2011*). In the standard set-up, moving the joystick from the center position to the left (CS+) is followed by a painful electrocutaneous stimulus (pain-US), whereas moving the joystick from the center position to the right (CS−) is not (counterbalanced across participants). This means that the endpoint location of the CS+ movement (i.e., the left side in the three-dimensional space with the body midline as reference) is an additional predictor of the pain-US; the endpoint of CS− movement is located on the opposite side and therefore the endpoint location is clearly different for both movements. In a between-subjects design, we added a group to the standard set-up with overlapping spatiotopic and proprioceptive information relating to the pain-US prediction. More specifically, participants moved the joystick from the left to the center (i.e., right movement) and from the right to the center (i.e., left movement); the center position being the endpoint location for both CS movements. The endpoint location in this newly added group is ambiguous, and can be conceptualized as an overlapping feature. In line with the feature overlap hypothesis, we hypothesized greater differential learning in the *location* group (i.e., endpoint location non-overlap for CS+ and CS−) than in the *movement* group (i.e., endpoint location overlap for CS+ and CS−). Subsequently, groups went through a crossover phase testing fear generalization; based on the principle of stimulus generalization, we hypothesized that if associative strength accrues to the proprioceptive features, generalization to novel, proprioceptively similar movements with different endpoint location would occur.

## METHODS AND MATERIALS

### Participants

A total of 51 healthy, pain-free indiviuals volunteered to participate in this study. The sample size was informed by previous research using a similar paradigm and a between-subjects design (*Geschwind et al., 2015*; *Meulders, Jans & Vlaeyen, 2015*). Volunteers were recruited through flyers and were paid €8 for their participation. Exclusion criteria were: pregnancy, current or history of cardiovascular disease, chronic or acute respiratory disease (e.g., asthma, bronchitis), neurological diseases (e.g., epilepsy), any current or past psychiatric disorder including clinical depression and panic/anxiety disorder, chronic pain, uncorrected hearing problems, painful wrist/hand, or related problems, cardiac pacemaker or the presence of any other electronic medical devices, and the presence of any other severe medical conditions. After checking these exclusion criteria

**Table 1 Demographic characteristics and scores on the psychological trait questionnaires for the location and the movement group separately.**

| Total $N = 51$ | Location group ($n = 26$) | | Movement group ($n = 25$) | | $t$ | $p$-value |
|---|---|---|---|---|---|---|
| | $M$ | $SD$ | $M$ | $SD$ | | |
| Age (in years) | 24.73 | 8.35 | 25.88 | 8.85 | −0.48 | 0.64 |
| Self-reported pain intensity (1–10) | 7.56 | 0.86 | 7.48 | 0.99 | 0.30 | 0.77 |
| Pain intensity (in mA) | 28.23 | 21.91 | 21.24 | 11.41 | 1.42 | 0.16 |
| PANAS—negative affect | 18.54 | 5.55 | 18.92 | 1.05 | −0.25 | 0.81 |
| PANAS—positive affect | 35.58 | 4.67 | 34.76 | 4.97 | 0.60 | 0.55 |
| STAI-T—total score | 37.69 | 10.86 | 37.60 | 9.04 | 0.03 | 0.97 |
| PCS—total score | 14.46 | 9.17 | 16.72 | 8.02 | −0.93 | 0.35 |
| PCS—magnification | 2.69 | 2.28 | 3.08 | 2.29 | −0.61 | 0.55 |
| PCS—rumination | 7.23 | 4.26 | 8.08 | 4.04 | −0.73 | 0.47 |
| PCS—helplessness | 4.54 | 3.72 | 5.56 | 3.19 | −1.05 | 0.30 |
| FPQ—total score | 60.19 | 15.75 | 64.12 | 13.14 | −0.97 | 0.34 |
| FPQ—medical pain | 19.46 | 7.17 | 23.88 | 6.21 | −2.35 | 0.02* |
| FPQ—minor pain | 16.62 | 5.83 | 17.72 | 5.50 | −0.70 | 0.49 |
| FPQ—severe pain | 24.12 | 6.46 | 22.52 | 7.67 | 0.80 | 0.43 |

Notes:
PANAS, Positive and Negative Affect Schedule; PCS, Pain Catastrophizing Scale; FPQ, Fear of Pain Questionnaire; STAI-T, trait portion of the State-Trait Anxiety Inventory; $M$, mean; $SD$, standard deviation.
* $p < 0.05$.

through self-report, elegible participants provided informed consent. The informed consent form emphasized that participation was completely voluntary and that participants were allowed to decline participation at any time during the experiment. Both groups did not differ on any psychological traits (fear of pain, pain catastrophizing, trait anxiety, and positive and negative affect) that may potentially affect the acquisition and generalization of fear of movement-related pain (see Table 1), except on the medical pain subscale of the Fear of Pain Questionnaire (FPQ) (*McNeil & Rainwater, 1998*; *Roelofs et al., 2005*): participants in the movement group (mean ± SD = 23.88 ± 6.21) scored higher than participants in the location group (mean ± SD = 19.46 ± 7.17), $t(49) = −2.35$, $p < 0.05$[1]. The experimental protocol was approved by the Social and Societal Ethics committee of the KU Leuven (registration number: S55434) and the Medical Ethical Committee of the University Hospital of the KU Leuven (registration number: ML9403).

[1] Please note that additional analyses including the medical pain subscale of the FPQ as a covariate did not change the results, suggesting that the observed differences between the groups are not due to the higher score on this index in the movement group.

## Stimulus material

The experiment was run on a Windows XP computer (Dell Optiplex 755) with 2 GB RAM and an Intel Core2 Duo processor at 2.33 GHz and an ATI Radeon 2400 graphics card with 256 MB of video RAM, using Affect 4.0 (*Spruyt et al., 2010*). Four proprioceptive stimuli, that is, moving a hydraulic force feedback Paccus Hawk joystick (Paccus Interfaces BV, Almere, The Netherlands) 90° to the left or right from different starting positions were used: Two of these stimuli served as CSs, for example, moving from the middle

position to the left and from the middle position to the right, and the other two stimuli served as generalization stimuli (GSs), for example, moving from the right to the middle position (i.e., moving the joystick to the left) and moving from the left to the middle position (i.e., moving the joystick to the right). Which stimuli served as CSs or GSs depended on group allocation, and which stimulus served as the CS+ and the CS− was counterbalanced across participants. An electrocutaneous stimulus of two ms duration serving as the painful unconditioned stimulus (pain-US) was delivered by a commercial stimulator (DS7A, Digitimer, Welwyn Garden City, England) through surface SensorMedics electrodes (eight mm diameter) filled with K–Y gel that were attached to the wrist of the dominant hand. The participants moved the joystick with their dominant hand, and the pain-US was delivered at the wrist of the same hand. During the calibration procedure the participants were repeatedly exposed to electrocutaneous stimuli of increasing intensity. They were asked to rate each stimulus on a scale from 0 to 10, with "0" indicating that they felt nothing, "1" indicating that they first felt the stimulation but that it was not aversive (sensory detection threshold), "2" indicating that the sensation was starting to become slightly painful (pain threshold), and "10" indicating the worst pain that they could imagine. A subjective stimulus intensity of "8" which refers to a stimulus that is "*significantly painful and demanding some effort to tolerate*" was targeted. When this stimulus was selected, the experimenter asked the participant whether s/he would agree to repeatedly receive stimuli of maximally this amplitude during the experiment.

## Procedure

We used an adapted version of the VJMP developed by Meulders, Vansteenwegen, and Vlaeyen (*Meulders, Vansteenwegen & Vlaeyen, 2011*) eliminating as much visual information as possible from the task. Contrary to the standard procedure we did (1) not include the vertical counter bars and thus no visual feedback about the number of movements that remained to be carried out was provided, and (2) use a signaled movement set-up in which an auditory cue (presented either in the left or right ear monaurally) indicated the movement direction. The experiment was conducted during a 60-min session and consisted of a preparation phase, a practice phase, a startle habituation phase, a fear acquisition phase, and a fear generalization phase. A between-subjects design was employed (see Table 2), 26/51, participants were randomly assigned to the location group and moved the joystick from the middle position to the left and to the right, and 25/51 participants were assigned to the movement group and moved the joystick from the left to the middle position and from the right to the middle position. A random number list generated using Excel was employed for the randomization. In both groups, one movement (CS+) was consistently followed by the pain-US and the other movement (CS−) was never followed by the pain-US during acquisition. In the location group not only the movement (i.e., proprioceptive information) but also the endpoint location of a movement (i.e., left or right in the three-dimensional space with the body midline as reference) served as an additional predictor (i.e., spatial information) for the occurrence of the pain-US. In the movement group, the endpoint location of every movement direction was the same (i.e., the centered position of the joystick) and thus remained ambiguous

**Table 2 Experimental design summary.**

| Group (N = 51) | Practice 2 × 8 trials | Startle habituation eight trials | Fear acquisition 4 × 8 trials | Fear generalization eight trials |
|---|---|---|---|---|
| Location Group (n = 26) | 4 × CS+ only | Eight startle probes | 4 × 4 CS+ | 4 GS+ |
| | 4 × CS− | | 4 × 4 CS− | 4 GS− |
| Movement Group (n = 25) | 4 × GS+ | | | |
| | 4 × GS− | | | |

Notes:
CS, conditioned stimulus; GS, generalization stimulus; CS+, joystick movement followed by the painful electrocutaneous stimulus (pain-US); CS−, joystick movement never followed by the pain-US; GS+, joystick movement in the same direction as the CS+ but from a different starting position; GS−, joystick movement in the same direction of the CS− but from a different starting position.
The suffix "only" indicates that a certain reinforced movement is not followed by the pain-US. GSs are never followed by the pain-US. In the Movement Group, CSs were movements from the middle position to the left and to the right, and GSs were movements from the left and the right to the middle position, but in the Location Group, opposite stimulus functions applied. During the acquisition and generalization phase a startle probe was presented on each trial: two startles probes were presented during the CS+ (GS+) and the CS− (GS−), and four during the intertrial interval.

with respect to the occurrence of the pain-US. Which joystick movements served as the CS+ and the CS− was counterbalanced across participants. During the generalization phase, the GSs in the movement group consisted of the CSs of the location group and the GSs in the location group consisted of the CSs in the movement group.

### Preparation phase

Participants were informed both by oral and written means that painful electrocutaneous stimuli (pain-USs), tones indicating the direction of the movements and short loud noises (acoustic startle probes) would be administered during the experiment. After the participants provided informed consent, the electrodes for the eyeblink startle responses were attached. Next, the stimulation electrodes were attached and the intensity level of the pain-US was individually selected (see calibration procedure; Section "Stimulus material").

### Startle habituation phase

A habituation phase was designed to prevent possible confounds in the data because startle responses to the first probes are usually relatively large. This phase contained eight trials, each lasting 17s. During each trial, one startle probe was presented (100 dBA burst of white noise) randomly, either between the 8th and 12th second or between the 13th and 17th second after trial-onset. The timing of probes during the trials were randomized across participants. After the last probe presentation a 2-s intertrial interval (ITI) was inserted before moving on to the next phase. During this phase, the headphones were put on and the central lighting of the experimental room was turned off. Small lamps provided dimmed light. Note that during this phase, no pain-USs were delivered.

### Practice phase

Before the practice phase started, we calibrated the range of motion of the joystick and participants received detailed instructions about the experimental task. They were told that at the beginning of each trial, a tone (i.e., the direction signal) would be presented either in their left or right ear and that their task was to move the joystick 90° in the direction

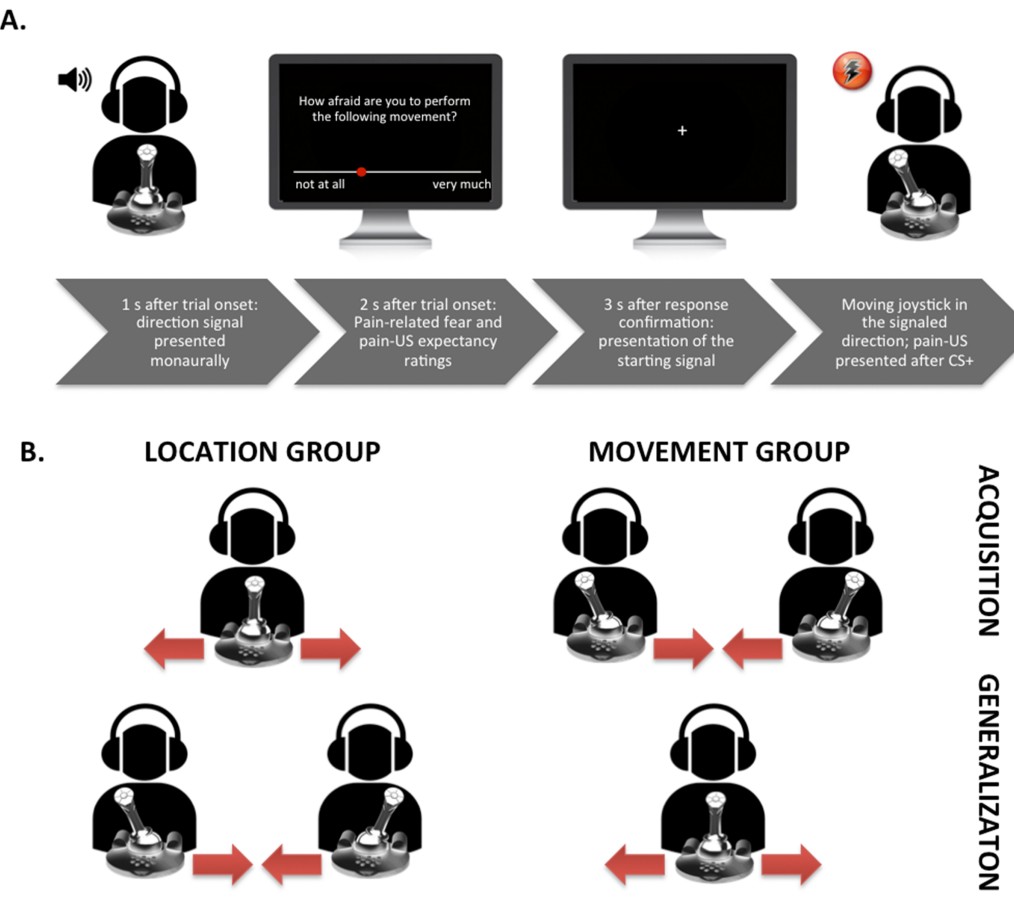

**Figure 1 Visual representation of the set-up.** (A) Exemplary trial timing in the location group; participant moving the joystick from the center to the left. (B) Visualization of the movements performed in the different groups during the acquisition and the generalization phases.

that the tone was presented. Participants were requested to move the joystick as quickly and accurately as possible when prompted by a starting signal "+" (fixation cross presented in the middle of the computer screen). In total, two blocks of eight trials were run: the first block comprised eight trials of the typical location group training, that is, participants performed four movements from the middle position to the left and four movements from the middle position to the right (see Fig. 1 for a schematic overview of the experimental task). The second block comprised eight trials of the typical movement group training, that is, participants moved four times from the left to the middle position and four times from the right to the middle position. Before each trial, the joystick was automatically reset to the appropriate starting position (the middle position (90° upright), the left (0°), or the right (180°), in accordance with the type of training the participant was receiving). Thus, in the first block a tone was presented either in the right ear or the left ear and the participants were instructed to move the joystick from the middle to the corresponding direction (again, only after the starting signal "+" was shown on the screen). Hence, if the tone was presented in the right ear, the participants had to move the joystick 90° to the right; likewise, if the tone was presented in the left ear, they had to move

the joystick 90° to the left. In the second block, a tone presented either in the left or right ear again indicated the movement direction and participants again were asked to move the joystick in the corresponding direction. However, this time, the original position of the joystick was either on the left (0°) or on the right (180°). This meant that if the joystick was on the left in its starting position, it should be moved 90° to the right (i.e., in the middle position) when there was a tone presented in the right ear; if the joystick was on the right in its starting position, it should be moved 90° to the left (i.e., in the middle position) when there was a tone presented in the left ear. In order to keep all procedural aspects consistent across both experimental groups (except for the crucial between-group manipulation), and not to provide the location group with an additional predictor for the occurrence of the pain-US, the tone was also presented in the movement group. The order of the blocks during practice was randomized between participants and the presentation of the CSs during each block were semi-randomized with the restriction that no more than two consecutive trials could be of the same type. During the practice phase participants received verbal feedback straightaway to learn to operate the joystick accurately. No pain-USs nor startle probes were delivered during the practice phase.

### Fear acquisition

This phase was basically the same as the practice phase, but now (1) pain-USs and startle probes were presented, (2) four blocks of one training type (movement or location group training) were run instead of two training blocks (one of each training type), and (3) instructions emphasized to pay close attention to the starting signal "+" and to respond as fast and accurately as possible upon its presentation.

On each trial, a tone was presented either in the participants' left or right ear 1 s after trial onset. When a trial consisted of prospective questions (see Fig. 1A), the rating scale was depicted on the screen 2 s after trial onset. When the participants had answered the questions, the joystick automatically reset to the proper position, and the starting signal "+" appeared 3 s later. Although the duration of the CS movement was of variable length depending on the participants' movement speed, each trial included a post-CS-ITI of 5 s. In each block of eight CS movements, four of the startle probes were presented during the CS movements, two during the CS+ and two during the CS– (200 ms after movement onset), and four during the ITI (randomly between 2.5 and 3.5 s after the CS movement was terminated). The pain-US occurred immediately after the CS+ movement was successfully terminated. CS+ and CS– trials per block were presented in a semi-randomized order with the restriction that no more than two consecutive trials could be of the same trial type. During each block fear of movement-related pain and the expectancy of the pain-US were measured. After each block the pain intensity and the unpleasantness of the pain-US were measured, as well the unpleasantness of the CSs.

### Fear generalization

In order to test whether participants transferred their knowledge about the stimulus contingencies acquired in the acquisition blocks, a generalization phase was included.

During this phase one block of eight trials was run and no pain-USs were presented after the GSs. The randomization schedule of the trials and the startle probe presentation per block was the same as in the previous phase. In this phase the movement direction (to the left or to the right) remained the same, while the starting point and the endpoint location of the movements changed. That is, participants in the location group, who moved the joystick in the acquisition blocks from de middle to the left and to the right, now moved the joystick in the generalization block from the left and right to the middle. Participants in the movement group, who moved the joystick in the acquisition blocks from the left and the right to the middle, now moved the joystick from the middle to the left and to the right.

## Primary outcome variables

### Prospective pain-US expectancy and self-reported fear of movement-related pain

During each block, pain-US expectancy ratings and fear ratings were collected once per movement type during acquisition and twice during the generalization phase (referred to as the "first generalization test" and the "second generalization test"). Participants answered the following questions: (1) *"To what extent do you expect an electrocutaneous stimulus after performing the following movement?"*, and (2) *"To what extent are you afraid to perform the following movement?"* on a numerical rating scale ranging from 0 to 10 with anchors "*not at all*" to "*very much*". Participants were instructed to browse through the scales with the joystick using a wrist rotation (i.e., the Z-rotation function of the Paccus Hawk joystick) and to click on the shooting button of the joystick to confirm their answer. The Z-rotation was used to avoid possible contamination with the CS/GS movements during the experiment.

### Eyeblink startle modulation

The eyeblink startle response is a component of the reflexive cross-species, full-body defensive response mobilization, which is triggered by startle-evoking stimuli (e.g., acoustic startle probe) and can be measured by the tension in the muscles underneath the eye. Startle modulation refers to the potentiation of the startle reflex during fear states elicited by the anticipation of an aversive stimulus (e.g., an electrocutaneous stimulus). Orbicularis Oculi electromyographic activity (EMG) was recorded with three Ag/AgCl Sensormedics electrodes (four mm) filled with electrolyte gel. After cleaning the skin with exfoliating peeling cream to reduce inter-electrode resistance, electrodes were placed on the left side of the face according to the site specifications proposed by *Blumenthal et al. (2005)*. The raw signal was amplified by a Coulbourn isolated bioamplifier with bandpass filter (LabLinc v75–04). The recording bandwidth of the EMG signal was between 90 Hz and 1 kHz (±3 dB). The signal was rectified online and smoothed by a Coulbourn multifunction integrator (LabLinc v76–23 A) with a time constant of 20 ms. The EMG signal was digitized at 1,000 Hz from 200 ms before the onset of the auditory startle probe until 1,000 ms after probe onset. The startle probe was a 100 dBA burst of white noise with instantaneous rise time presented binaurally for 50 ms through headphones (Philips SHP2500).

Eyeblink startle responses elicited by startle probes delivered during the CS movements served as an index of cued pain-related fear. Eyeblink startle responses elicited by startle probes during the ITI served as an index of contextual pain-related fear (*Vansteenwegen et al., 2008*).

## Secondary outcome variables

### Retrospective unpleasantness of the CS/GS movements

After each block the following question was presented: *"How unpleasant did you find moving to the left/right?"* Responses were indicated on an 11-point numerical rating scale ranging from 0 to 10 with 0 meaning *"not at all"* and 10 meaning *"very much."*

### Retrospective pain-US intensity and unpleasantness

After each acquisition block, the following questions were asked: (1) *"How painful did you find the electrocutaneous stimuli in the previous block?"* and (2) *"How unpleasant did you find the electrocutaneous stimuli in the previous block?"* Responses were indicated on an 11-point numerical rating scale ranging from 0 to 10 with 0 meaning *"not at all"* and 10 meaning *"very much."*

### Psychological trait questionnaires

After the experiment the participants filled in a set of questionnaires to map out individual differences for descriptive purposes: the Dutch version of the FPQ (*McNeil & Rainwater, 1998*; *Roelofs et al., 2005*), the Dutch version of the Pain Catastrophizing Scale (*Van Damme et al., 2002*; *Sullivan, Bishop & Pivik, 1995*), the Dutch version of the Positive and Negative Affect Schedule (*Engelen et al., 2006*; *Watson, Clark & Tellegen, 1988*), and the Dutch version of trait version of the State-Trait Anxiety Inventory (*Spielberger, 1983*; *Van der Ploeg, 2000*).

## Experimental setting

Participants were seated in an armchair (0.6 m screen distance) in a sound-attenuated and dimmed experimental room, adjacent to the experimenter's room. Further verbal communication was possible through an intercom system; the experimenter observed the participants and their physiological responses online by means of a closed-circuit TV installation and computer monitors.

## Response definition and data analysis overview

Using PSPHA (*De Clercq et al., 2006*), a modular script-based program, we calculated the peak amplitudes defined as the maximum of the response curve within 21–175 ms after the startle probe onset. All startle waveforms were visually inspected off-line, and technical abnormalities and artifacts were eliminated using the PSPHA software. Every peak amplitude was scored by subtracting its baseline score (averaged EMG level between 1 and 20 ms after the probe onset). The raw scores were transformed to $z$-scores to account for inter-individual differences and T-scores were used to visualize the data. Averages were calculated for responding during CS/GS movements and ITI separately for the movement and the location group. Participants who failed to show an elevated peak amplitude

compared to baseline on more than 30% of probe trials were considered non-responders and were excluded from statistical analyses. A total of four participants from our sample were excluded due to the absence of reliable startle eyeblink responses, therefore the statistical analyses of the psychophysiological data were run on a total sample of 47 participants.

Repeated measures (RM) ANOVAs were run to examine differences in acquisition and generalization between the movement group and the location group for each dependent variable. These analyses included Group as a between-subject variable, and Stimulus Type and Block as within-subject variables. Because we had clear a priori hypotheses, data were further analyzed using planned comparisons. Significance levels were set at $\alpha = 0.05$, effect size $\eta_p^2$ is reported for significant effects. Greenhouse-Geisser corrections were applied when the sphericity assumption was violated; uncorrected degrees of freedom are reported together with $\varepsilon$ and corrected $p$-values. All statistical analyses were carried out with Statistica 13 software (StatSoft, Inc., Tulsa, OK, USA).

## RESULTS

### Pain-US characteristics

Although participants in the location group tended to select a more intense electrocutaneous stimulus (mean $\pm$ SD = 28.23 $\pm$ 21.91 mA) than participants in the movement group (mean $\pm$ SD = 21.24 $\pm$ 11.41 mA), this difference was not significant, $t(49) = 1.42$, $p = 0.16$. The self-reported pain intensity of the electrocutaneous stimulus in the location (mean $\pm$ SD = 7.56 $\pm$ 0.86) and movement (mean $\pm$ SD 7.48 $\pm$ 0.99) group also did not differ significantly, $t(49) = 0.30$, $p = 0.77$.

### Primary outcome variables

#### Prospective pain-US expectancy

Figure 2 displays the mean prospective pain-US expectancy ratings for both the location and the movement group separately for the five blocks of the experiment (four ratings during acquisition, one per block and two ratings during one generalization block). A $2 \times 2 \times 6$ (Group (location/movement) $\times$ Stimulus Type (CS+/CS−) $\times$ Block (A1–A4, G1–G2)) RM ANOVA revealed a significant main effect of Stimulus Type, $F(1, 49) = 109.08$, $p < 0.001$, $\eta_p^2 = 0.69$, and a significant main effect of Block, $F(5, 245) = 5.21$, $p < 0.01$, $\varepsilon = 0.64$, $\eta_p^2 = 0.10$. The main effect of Group did not reach significance, $F = 0.04$, $p = 0.85$. Interestingly, there was a significant Stimulus Type $\times$ Group interaction, $F(1, 49) = 9.98$, $p < 0.01$, $\eta_p^2 = 0.17$, indicating that the difference in pain-US expectancy ratings between the CS+ and CS− differed in both groups. Also the Stimulus Type $\times$ Block interaction turned out to be significant, $F(5, 245) = 18.82$, $p < 0.001$, $\varepsilon = 0.76$, $\eta_p^2 = 0.28$, indicating that the difference in pain-US expectancy for the CS+ versus CS− grew larger over time. This two-way interaction was modulated by Group, $F(5, 245) = 3.06$, $p < 0.05$, $\varepsilon = 0.76$, $\eta_p^2 = 0.06$, suggesting that the differences in pain-US expectancy between the CSs gradually grew larger across blocks, and this increase was more substantial in the location group compared to the movement group.

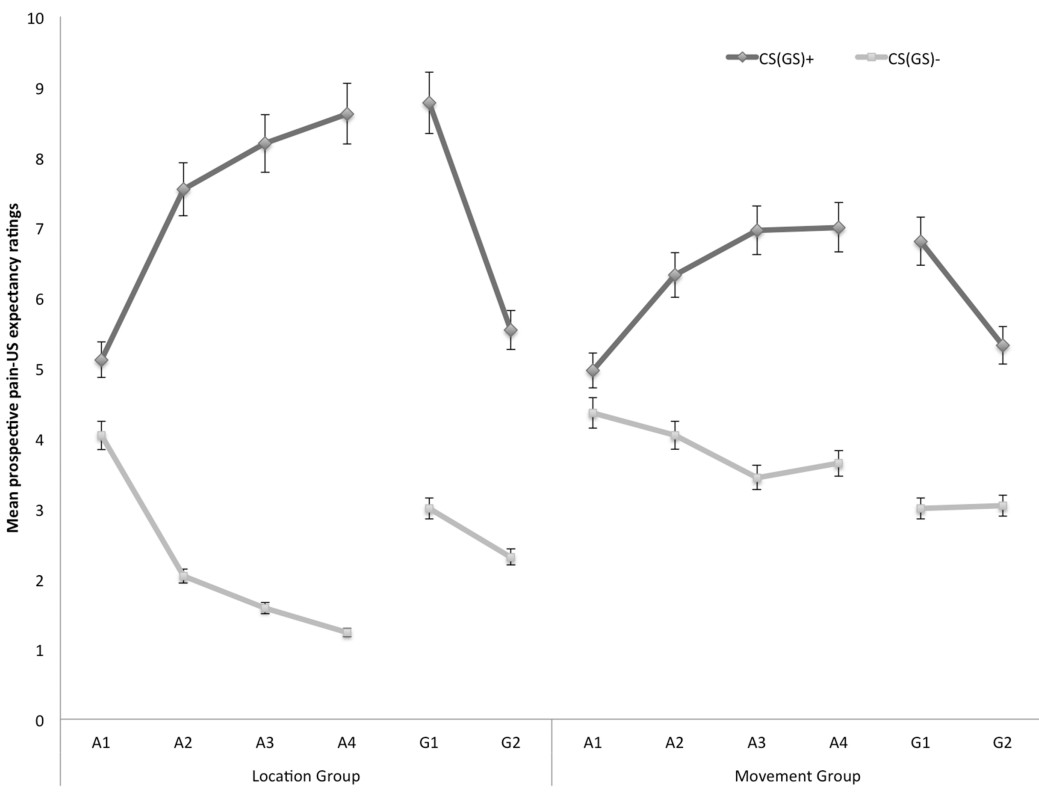

**Figure 2** Mean prospective pain-US expectancy ratings during the four acquisition blocks (A1–A4) and the generalization block (G1–2) for the location group and the movement group separately. Error bars denote 95% confidence intervals.

*Acquisition.* Planned comparisons at the beginning of the acquisition phase (A1), revealed no differences in pain-US expectancy for the CS+ and CS− movements in the location group, $F(1, 49) = 3.23$, $p = 0.08$, nor the movement group, $F(1, 49) = 0.96$, $p = 0.33$. Yet, at the end of the acquisition (A4) participants in the location group gave higher pain-US expectancy ratings for the CS+ movements than for the CS− movements, $F(1, 49) = 99.55$, $p < 0.001$. Likewise, participants in the movement group expected the pain-US to occur more when they performed the CS+ movement than when performing the CS− movement, $F(1, 49) = 19.82$, $p < 0.001$. A between-group contrast further showed that the difference in pain-US expectancy between the CS+ and the CS− was significantly larger in the location than in the movement group, $F(1, 49) = 14.49$, $p < 0.001$.

*Generalization.* To examine the generalization effect, within-group planned comparisons were conducted showing that participants in the location group, generalized the acquired differential pain-US expectancy ratings to the first generalization test (G1), $F(1, 49) = 59.84$, $p < 0.001$, and the second generalization test (G2), $F(1, 49) = 21.25$, $p < 0.001$. Furthermore, planned comparisons showed that in the location group the difference in pain-US expectancy between the CS+ and the CS− was significantly smaller during generalization (G1) as compared to the end of the acquisition (A4), $F(1, 49) = 5.98$, $p = 0.02$, which seems to be due to an increase on the CS− instead of a decrement relating

to the CS+. In the movement group, the pain-US expectancy ratings were also higher when performing the CS+ movement than when performing the CS− movement in the first (G1), $F(1, 49) = 24.96$, $p < 0.001$, and second test of generalization (G2), $F(1, 49) = 10.18$, $p < 0.01$. No difference in differential pain-US expectancy ratings was observed between at the end of the acquisition (A4) compared to the first test of generalization (G1), $F(1, 49) = 0.43$, $p = 0.52$.

To further analyze the development of generalization, planned comparisons were performed between the end of the acquisition (A4) and the second generalization test (G2). In the location group, the difference in pain-US expectancy between the CS+ and the CS− was significantly smaller at G2 compared to A4, $F(1, 49) = 5.98$, $p = 0.02$, suggesting that extinction took place. Indeed, participants did not receive any pain-USs during the generalization phase. In the movement group, however, differential pain-US expectancy ratings for the CS+ and the CS− at G2 were not significantly reduced in comparison to A4, $F(1, 49) = 0.43$, $p = 0.52$.

### Prospective self-reported fear of movement-related pain

In Fig. 3 the mean prospective fear of movement-related pain ratings for both the location and the movement group are shown separately for the five blocks of the experiment (four ratings during acquisition, one per block, and two ratings during one generalization block). A $2 \times 2 \times 6$ (Group (location/movement) $\times$ Stimulus Type (CS+/CS−) $\times$ Block (A1–4, G1–2)) RM ANOVA revealed significant main effects of Stimulus Type, $F(1, 49) = 60.74$, $p < 0.001$, $\eta_p^2 = 0.55$, and Block, $F(5, 245) = 4.84$, $p < 0.001$, $\varepsilon = 0.78$, $\eta_p^2 = 0.09$. The main effect of Group was not significant, $F(1, 49) = 0.13$, $p = 0.72$. The Stimulus Type $\times$ Group interaction turned out to be significant, $F(1, 49) = 7.32$, $p < 0.01$, $\eta_p^2 = 0.13$, indicating that differences in fear of movement-related pain between the CS+ and the CS− varied depending on the Group. Also the Stimulus Type $\times$ Block interaction turned out to be significant, $F(5, 245) = 13.78$, $p < 0.01$, $\varepsilon = 0.66$, $\eta_p^2 = 0.22$, indicating that difference in fear of movement-related pain elicited by the CS+ and CS− grew larger over blocks. Although this two-way interaction was not modulated by Group, $F(5, 245) = 1.79$, $p = 0.15$, $\varepsilon = 0.66$, we continued testing our a priori hypotheses.

*Acquisition.* Planned comparisons at the beginning of the acquisition phase (A1) revealed no differences in fear reported in response to CS+ and CS− movements in the location group, $F(1, 49) = 1.08$, $p = 0.30$, nor the movement group, $F(1, 49) = 0.05$, $p = 0.83$. At the end of the acquisition phase (A4) however, participants were more afraid of the CS+ than the CS− both in the location group, $F(1, 49) = 49.29$, $p < 0.001$, and in the movement group, $F(1, 49) = 12.82$, $p < 0.001$. Planned between-group comparisons at A4 confirmed that differential fear of movement-related pain was more pronounced in the location group than in the movement group, $F(1, 49) = 5.57$, $p = 0.02$.

*Generalization.* To examine the fear generalization effect, within-group planned comparisons were conducted showing that participants in the location group, generalized the acquired differential fear of movement-related pain to the first generalization test (G1), $F(1, 49) = 26.73$, $p < 0.01$, and to the second generalization test (G2), $F(1, 49) = 10.97$, $p < 0.01$. Furthermore, the difference in fear of movement-related pain reported for the

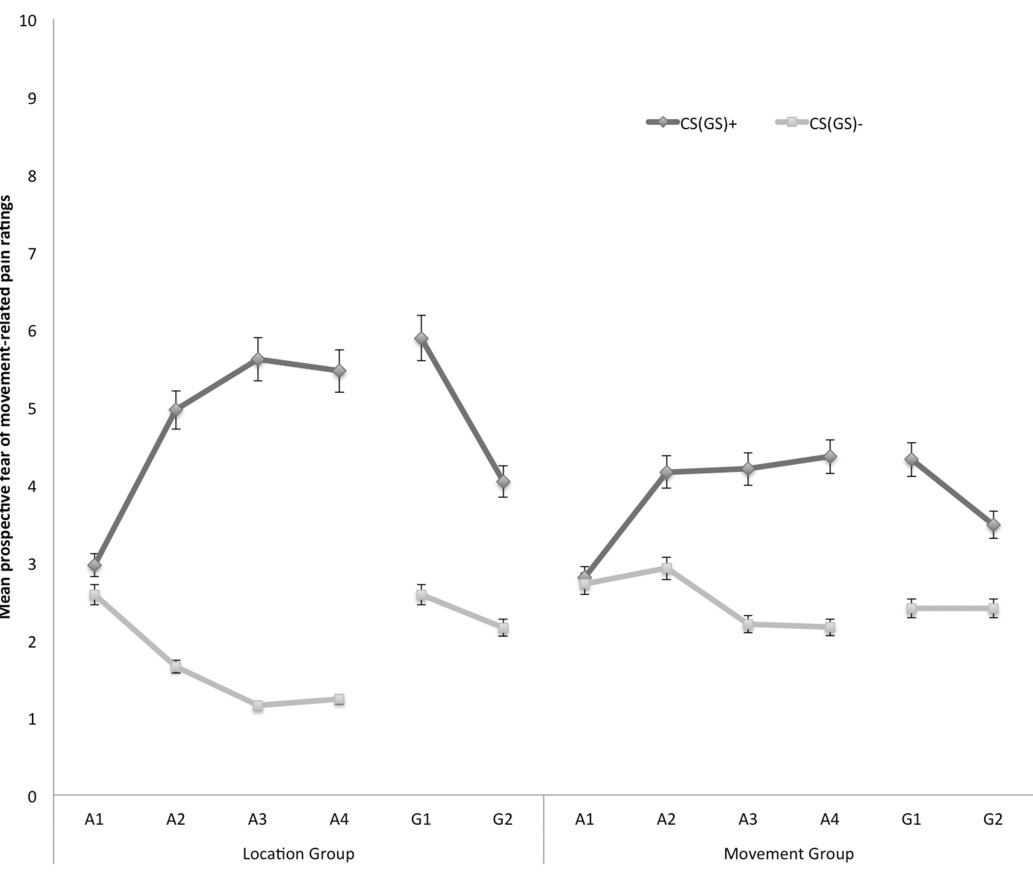

**Figure 3 Mean prospective fear of movement-related pain ratings during the four acquisition blocks (A1–A4) and the generalization block (G1–2) for the location group and the movement group separately.** Error bars denote 95% confidence intervals. 

CS+ and the CS− tended to be smaller during generalization (G1) as compared to the end of the acquisition (A4) in the location group, $F(1, 49) = 3.86$, $p = 0.06$, which seems to be due to an increase on the CS− instead of a decrement relating to the CS+. In the movement group, the fear of movement-related pain ratings were also higher when performing the CS+ movement than when performing the CS− movement in the first (G1), $F(1, 49) = 8.66$, $p < 0.01$. At the second test of generalization (G2), $F(1, 49) = 3.46$, $p = 0.07$, this difference was only borderline significant, suggesting that extinction took place. No difference in differential fear ratings was observed between the end of the acquisition and the first test of generalization in the movement group, $F(1, 49) = 0.34$, $p = 0.56$.

To further analyze the development of generalization, planned comparisons were performed between the end of the acquisition (A4) and the second generalization test (G2). In the location group, the difference in fear of movement-related pain between the CS+ and the CS− was significantly smaller at G2 compared to A4, $F(1, 49) = 14.97$, $p < 0.01$, suggesting that extinction took place. Note that participants indeed did not receive any pain-USs during the generalization phase. In the movement group, however, the difference in fear of movement-related pain elicited by the CS+ and the CS− was not significantly reduced at G2 compared with A4, $F(1, 49) = 3.28$, $p = 0.08$.

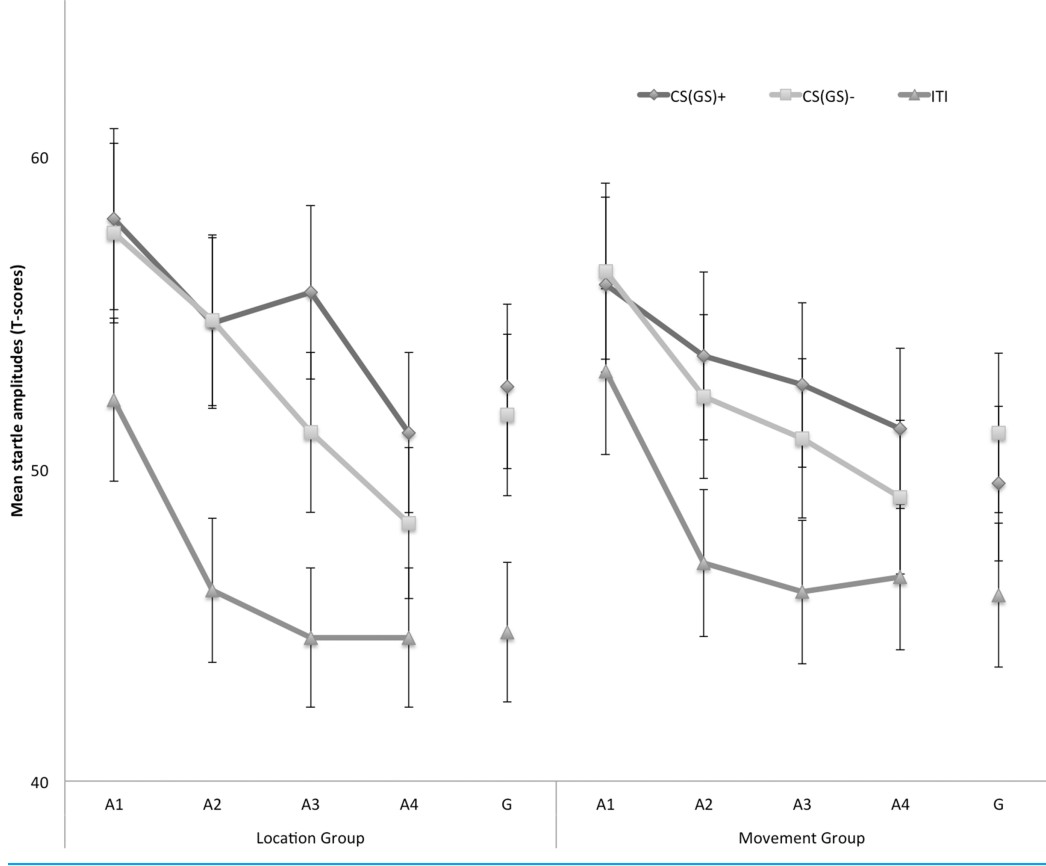

**Figure 4 Mean eyeblink startle amplitudes during the CS/GS movements and the ITI during the four acquisition blocks (A1–A4) and the generalization block (G) for the location and the movement group separately.** Error bars denote 95% confidence intervals.

### Eyeblink startle modulation

Figure 4 displays the fear-potentiated startle results for both the location and movement group separately for the five blocks of the experiment. The startle modulation data were analyzed with a $2 \times 3 \times 5$ (Group (location/movement) $\times$ Stimulus Type (CS+/CS−/ITI) $\times$ Block (A1–4, G)) RM ANOVA. This analysis revealed a significant main effect of Block, $F(4, 180) = 22.11$, $p < 0.001$, $\varepsilon = 0.84$, $\eta_p^2 = 0.33$, indicating an overall decrease in startle responding which is indicative of habituation. The main effect of Stimulus Type was also significant, $F(2, 90) = 40.77$, $p < 0.01$, $\eta_p^2 = 0.48$, but the main effect of Group was not, $F(1, 45) = 3.60$, $p = 0.06$. None of the interaction effects were significant, nonetheless we continued testing our a priori hypotheses.

*Acquisition.* Because startle responses elicited by few startle probes are less reliable, planned comparisons were carried out to evaluate differences in early (A1–A2) and late (A3–A4) acquisition. This analysis revealed that in the early acquisition participants did not demonstrate any differences in startle amplitudes in response to CS+ and CS− movements in the location group, $F(1, 45) = 0.01$, $p = 0.91$. In the late acquisition, however, startle amplitudes in the location group were higher in response to the CS+

then in response to the CS−, $F(1, 45) = 4.52$, $p = 0.04$. In the movement group, participants did not demonstrate differential startle amplitudes in the early acquisition, $F(1, 45) = 0.06$, $p = 0.80$, nor the late acquisition phase, $F(1, 45) = 1.12$, $p = 0.30$. Interestingly, during the late acquisition phase, startle amplitudes during the ITI were elevated in the movement group compared with the location group, $F(1, 45) = 4.85$, $p = 0.03$, suggesting that more fear of movement-related pain accrued to the context the movement group, whereas this effect was not present during early acquisition, $F(1, 45) = 0.85$, $p = 0.36$.

*Generalization.* To examine the generalization effect, planned within-group comparisons were performed during the test of generalization (G). In the location group, the acquired differential startle responding did not generalize to G1, $F(1, 45) = 0.37$, $p = 0.55$, this was largely due to inflated responses to the CS−. Because there was no acquisition effect in the movement group, we did not continue testing for generalization.

## Secondary outcome variables
### Retrospective unpleasantness of CS/GS movements
In Fig. 5, the mean retrospective unpleasantness ratings for both CS/GS movements during the four acquisition blocks and the generalization test are displayed separately for the location group and the movement group. A 2 × 2 × 5 (Group (location/movement) × Stimulus Type (CS+/CS−) × Block (A1–4, G1)) RM ANOVA yielded significant main effects of Stimulus Type, $F(1, 49) = 76.28$, $p < 0.001$, $\eta_p^2 = 0.61$, and Block, $F(4, 196) = 37.25$, $p < 0.001$, $\varepsilon = 0.71$, $\eta_p^2 = 0.43$. The main effect of Group did not reach significance, $F(1, 49) = 1.02$, $p = 0.32$. Interestingly, there was a significant Stimulus Type × Group interaction, $F(1, 49) = 5.00$, $p = 0.03$, $\eta_p^2 = 0.09$, indicating that differential retrospective unpleasantness ratings for CS+ and CS− depended on the Group. The Stimulus Type × Block interaction also turned out significant, $F(4, 196) = 25.03$, $p < 0.01$, $\varepsilon = 0.85$, $\eta_p^2 = 0.34$. None of the other interactions were significant.

*Acquisition.* Planned comparisons at the end of the acquisition phase (A4) confirmed that participants in both groups found the CS+ more unpleasant than the CS−, location group: $F(1, 49) = 39.16$, $p < 0.001$, movement group: $F(1, 49) = 19.33$, $p < 0.001$. There were no differences in differential unpleasantness reported for the CS+ and the CS− between both groups at A4, $F(1, 49) = 1.54$, $p = 0.22$.

*Generalization.* To examine whether unpleasantness of the CSs generalized to the GSs, planned comparisons were performed between the end of the acquisition (A4) and the generalization test (G1). In the location group the difference in retrospective unpleasantness ratings in response to the CS+ and the CS− was significantly smaller at G1 than at A4, $F(1, 49) = 26.77$, $p < 0.001$. A similar pattern occurred in the movement group, $F(1, 49) = 16.75$, $p < 0.001$. Further planned comparisons at G1 revealed that participants in the location group also rated the GS+ movement as more unpleasant than the GS− movement, $F(1, 49) = 6.54$, $p = 0.01$. However, in the movement group the differential unpleasantness ratings for the CS+ and the CS− did not generalize to the GSs, $F(1, 49) = 0.87$, $p = 0.36$.

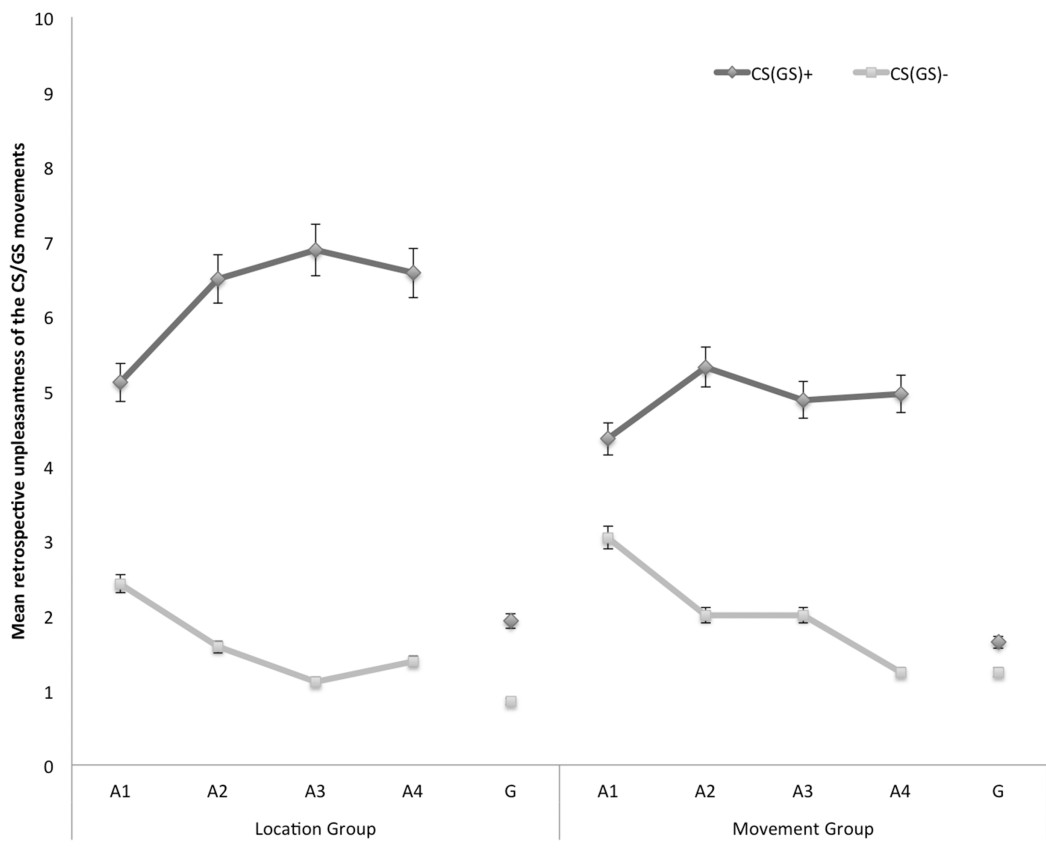

**Figure 5** **Mean retrospective unpleasantness of the CS/GS movements during the four acquisition blocks (A1–A4) and the generalization block (G) for the location and the movement group separately.** Error bars denote 95% confidence intervals.

### Pain-US intensity and unpleasantness

Figure 6 shows the mean ratings of pain intensity and unpleasantness of the pain-US for both groups separately for the four acquisition blocks. A $2 \times 2 \times 4$ (Group (location/ movement) $\times$ Rating (intensity/unpleasantness) $\times$ Block (A1–A4)) RM ANOVA yielded a significant main effect of Rating, $F(1, 49) = 19.29$, $p < 0.001$, $\eta_p^2 = 0.28$, indicating that, in general, ratings of unpleasantness were higher than pain intensity ratings. Although the figure seems to suggest that participants in the location group rated the pain-US as more painful and more unpleasant than the movement group, the main effect of Group was not significant, $F(1, 49) = 0.92$, $p = 0.34$. None of the other main effects or interactions reached significance.

## DISCUSSION

This study aimed to address the intriguing question as to what extent spatiotopic information contributes to the acquisition of fear of movement-related pain in addition to proprioceptive or movement-related information. First, we wanted to disentangle the contribution of proprioceptive and spatiotopic information relating to the prediction of the pain-US in the acquisition of fear of movement-related pain in the VJMP and assess whether fear of movement-related pain would generalize based on proprioceptive
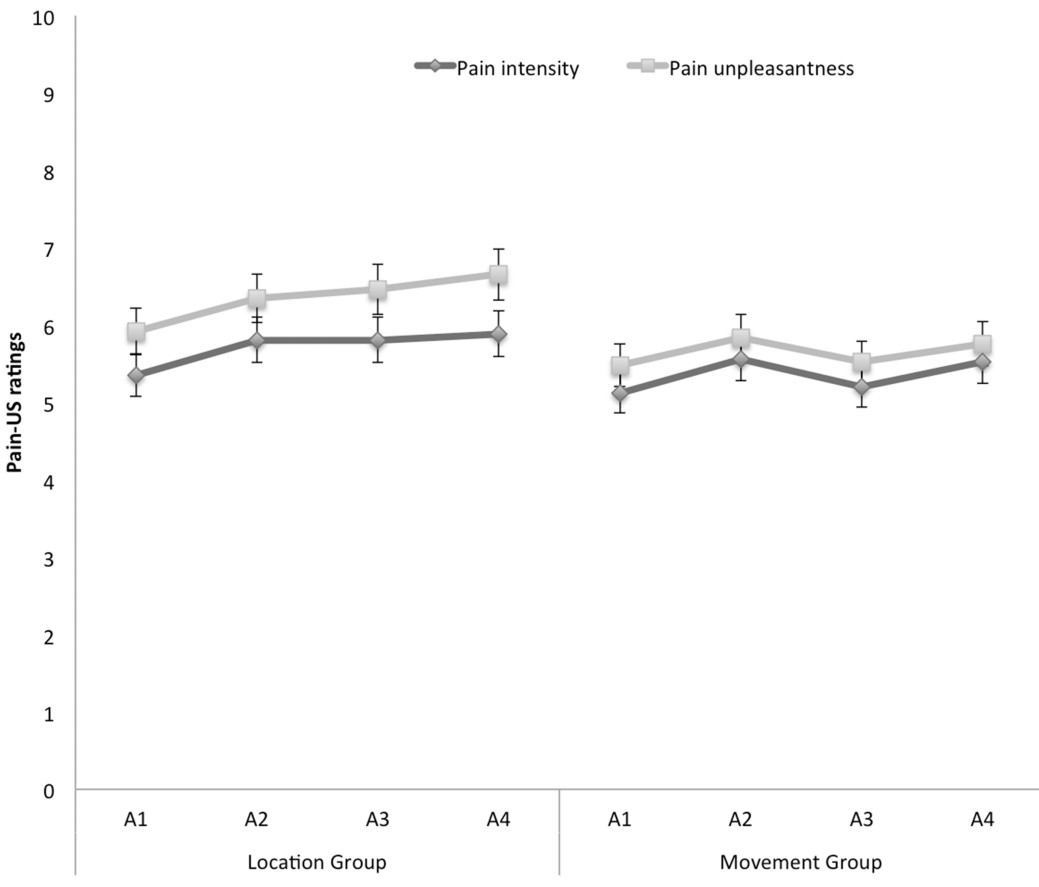

**Figure 6 Mean retrospective pain-US intensity and unpleasantness ratings for the location and the movement group during acquisition (A1–A4).** Error bars denote 95% confidence intervals.

information alone. In a between-subjects, crossover design, proprioceptive stimuli associated with joystick movements served as CSs and an electrocutaneous stimulus as the pain-US. In both groups, one movement (CS+) was consistently followed by the pain-US and another movement (CS−) was not. In the location group, participants moved the joystick from the middle position to the left and to the right; basically, the training in this group reflected the standard VJMP procedure. Participants in the movement group moved the joystick from the left and from the right to the middle position. The middle position being the endpoint location for both CS movements therefore became ambiguous with respect the pain-US occurrence. *Feature overlap* between the CS+ and CS− has been shown to reduce differential learning (*Haddad et al., 2012*). Therefore, we hypothesized greater differential learning in the location group (i.e., endpoint location non-overlap for CS+ and CS−) than in the movement group (i.e., endpoint location overlap for CS+ and CS−). Second, we tested whether generalization occurs to novel movements with similar proprioceptive features but with a different starting point and endpoint location (GSs).

We observed higher pain-US expectancies for the CS+ than the CS− in both the location and the movement group. In line with the feature overlap hypothesis, differential
acquisition in the pain-US expectancy ratings was greater in the location group than in the movement group. A similar data pattern was observed in the fear of movement-related pain ratings. That is, self-reported fear of the CS+ was higher than for the CS−, and this difference was larger in the location group than in the movement group. A more extreme pattern emerged in the startle eyeblink measures. Startle amplitudes at the end of acquisition were higher in response to the CS+ than to the CS− in the location group, but the difference between the CS+ and the CS− did not reach significance in the movement group. Interestingly, at the end of the acquisition phase, ITI startle amplitudes were elevated in the movement group compared with the location group, suggesting that in that group more fear accrued to the context. Taken together, these data provide strong support for the feature overlap hypothesis; demonstrating that differential fear of movement-related pain acquisition effects are larger when proprioceptive and spatiotopic features of the CS+ and the CS− do not overlap. With respect to the secondary outcome measures, visual inspection also suggested larger differences in unpleasantness ratings for the CS+ and CS− in the location group than in the movement group, statistical tests however failed to support this. At the end of the acquisition phase, the CS+ was found to be more unpleasant than the CS− in both groups.

Testing generalization of conditioned responses to novel movements with similar proprioceptive characteristics but with a different starting point and endpoint location revealed that differential pain-US expectancy ratings generalized in both groups. In line with the feature overlap hypothesis, there was a larger generalization decrement in the location group than in the movement group. Interestingly, this was due to an increase in pain-US expectancies in response to the GS− instead of a generalization decrement reflecting lower pain-US expectancies for the GS+ in the beginning of the generalization phase. Because no pain-USs were delivered, generalized differential responses in the location group were significantly reduced by the end of the generalization phase, but the difference between the GS+ and GS− remained statistically significant. In the movement group however, this extinction effect was less manifest, most likely because there was less differential learning to start with. A similar data pattern emerged in the fear of movement-related pain ratings: generalization occurred in both groups. In the line with the expectancy ratings, extinction of generalization at least partly occurred in both the location group and the movement group. In the startle responses, no generalization was observed in the location group, this effect again seems to be due to an increase in GS− responding but not to a decrement relating to the GS+. Because of the lack of differential learning during acquisition, we did not continue to test for generalization effects in the movement group. Taken together, these data suggest that proprioceptive information alone indeed can drive generalization effects; differential learning generalized at least in two of the three outcome variables.

In both the location and the movement group, the retrospective unpleasantness ratings show a large decrement in differential responding to the GSs, this is probably explained by generalization being tested under extinction and ratings collected at the end of the generalization phase. This is in contrast with the pain-US ratings and fear of movement-related pain ratings, which were collected prospectively, allowing for hindsight, experience-based corrections.

Some observations deserve more in-depth discussion. First, the ITI startle amplitudes were significantly higher in the movement group compared to the location group at the end of the acquisition phase, indicating that participants were more uncertain about the non-occurrence of the pain-US during the context in the movement group and thus were in general more fearful. This could be due to the ambiguity and feature overlap between the CS+ and CS− present in the movement group. It is plausible that feature overlap did not only lead to reduced differential learning, but also created some degree of unpredictability which increased contextual fear (*Grillon, 2002a*, *2002b*) in the movement group.

Second, the finding that generalization occurred to movements with similar proprioceptive features but another endpoint location confirms that the movement alone has gained sufficient associative strength to subsequently generate generalization effects. However, there was a generalization decrement in the location group, which seems to be due to an increase on the GS− instead of a decrement relating to the GS+. This increase in GS− responding may be explained as follows: during the generalization phase the participants in the location group lose the spatiotopic predictor for the pain-US. As a result, the CS− becomes especially "uncertain" or "ambiguous" because it now has a feature in common with the GS+. Participants therefore may not fully trust the GS−. Responses to the GS+ on the other hand seem to generalize perfectly when the spatiotopic predictor is lost, suggesting that when it comes to the threatening movement participants apply the "better safe than sorry" adagio. In the movement group on the contrary, the spatiotopic information was never a predictor of the pain-US in the first place and thus has not acquired any associative strength, therefore no such generalization decrement is observed in this group.

Some limitations should be considered as well. First, the joystick was programmed to automatically move into the proper starting position, especially in the movement group this may serve as an additional predictor for the pain-US. In addition, this set-up may render the task easier in this group and requiring less attention to the direction signal (i.e., tone presented monaurally), because the joystick is physically limited to move in only one direction. Therefore, in the movement group, the starting point (i.e., left or right) may also gain associative strength. However, based on a componential CS representation approach, late components in a (chain of) CS(s) typically gain excitatory properties by virtue of their temporal proximity with the US, whereas early components in a (chain of) CS(s) gain inhibitory properties, referred to as the law of inhibition of delay (*Vogel, Brandon & Wagner, 2003*). This means that in our study participants will learn that the pain-US will not occur (i.e., inhibitory learning) when the joystick is positioned at the start location (early CS), but only when they actually performed the movement (late CS). A second limitation may be that we used a signalled movement set-up, in which a tone indicates the movement direction. This tone as well could serve as an additional predictor of the pain-US. Based on the same temporal proximity argument, however, it is highly unlikely that the direction signal gained more associative strength than the movement itself.

Although this is an experimental study in healthy participants investigating the basic processes underlying the acquisition and generalization of fear of movement-related pain,
we could speculate about the potential clinical implications of the current results. Previous research shows that exposure in multiple contexts (*Vansteenwegen et al., 2007*) and/or to variations of the feared object (*Rowe & Craske, 1998*) enhances the chance on generalization of extinction, which in turn prevents relapse. For example, a recent study in CRPS-I patients showed that increasing the number of different movements that patients are exposed to increases the chance of generalization of extinction, whereas repeated exposure to the same movements does not necessarily strengthen the generalization of extinction (*den Hollander et al., 2018*). Following this reasoning and if both spatiotopic and proprioceptive features of a movement/activity possess associative strength, it would be beneficial to expose chronic pain patients to movements/activities from different reference points in the three-dimensional space/with different start- and endpoints (e.g., when crossing arms).

To conclude, we successfully demonstrated that proprioceptive information is sufficient for the acquisition of fear of movement-related pain and serves a basis for the generalization of such fear at least in the verbal fear and pain-US expectancy measures. Furthermore, we showed that spatiotopic information in the standard VJMP also significantly contributes to the observed learning effects showing larger differential learning effects in the location group than in the movement group. These results further strengthen the construct validity of the VJMP and support its use to study learning processes involved in fear of movement-related pain.

## ACKNOWLEDGEMENTS

The authors would like to thank Lore Hulsmans for her assistance in the data collection.

### Funding

Ann Meulders is a postdoctoral researcher of the Research Foundation Flanders (FWO-Vlaanderen), Belgium, (grant ID 12E3717N) and is also supported by a Vidi grant from the Netherlands Organization for Scientific Research (NWO), The Netherlands (grant ID 452-17-002). Johan Vlaeyen is supported by the "Asthenes" long-term structural funding—Methusalem grant (# METH/15/011) by the Flemish Government, Belgium. The funders had no role in study design, data collection and analysis, decision to publish, or preparation of the manuscript.

### Grant Disclosures

The following grant information was disclosed by the authors:
Research Foundation Flanders (FWO-Vlaanderen), Belgium: 12E3717N.
Netherlands Organization for Scientific Research (NWO), The Netherlands: 452-17-002.
"Asthenes" long-term structural funding—Methusalem grant (# METH/15/011) by the Flemish Government, Belgium.

## Competing Interests

The authors declare that they have no competing interests.

## Author Contributions

- Ann Meulders conceived and designed the experiments, analyzed the data, contributed reagents/materials/analysis tools, prepared figures and/or tables, authored or reviewed drafts of the paper, approved the final draft.
- Johan W. Vlaeyen conceived and designed the experiments, approved the final draft.

## Human Ethics

The following information was supplied relating to ethical approvals (i.e., approving body and any reference numbers):

The KU Leuven Social and Societal Ethics Committee has approved the study protocol (reference number: S-55434).

## Data Availability

Datasets S1, S2, and S3 include the questionnaire data, startle eyeblink data, and self-reports. The legend in each dataset includes the explanation of the variable names.

## Supplemental Information

Supplemental information for this article can be found online at http://dx.doi.org/10.7717/peerj.6913#supplemental-information.

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
