# Peer review of "The effect of differential spatiotopic information on the acquisition and generalization of fear of movement-related pain"

_PeerJ, doi:10.7717/peerj.6913_

## Round 0.1 · original submission · Minor Revisions

Dear authors

Please find enclosed the review of your paper submitted to PeerJ. The revision take longer than I would expect due to some difficulties in finding reviewers. However, now we have important comments. I would like to invite you to address each comment, providing a response letters and a revised manuscript highlighting any change made to the manuscript.

·

Basic reporting

No comment.

Experimental design

No comment.

Validity of the findings

No comment.

Additional comments

The present study investigated to what extent spatiotopic information contributes to the acquisition and generalization of fear besides proprioception. Based on controled joystick movements, conditioned stimulus associated with pain and conditioned stimulus not associated with pain were used to understand differential learning.

It has been shown that pain-related fear might be more disabling than pain itself, specially for chronic mausculoskeletal pain patients. The present study tested pain fear generalization to movements with similar proprioceptive features but different endpoint location. The authors suggest that fear accrued to proprioceptive conditioning stimulus features.

This is an interesting study that can have a positive impact on the understanding of sensory-motor interactions. In my opinion, in general, the manuscript is clear, unambiguous, well-written, and structured according with PeerJ standards. Literature is well referenced and relevant. Raw data is supplied. The investigation has been conducted in conformity with the prevailing ethical standards in the field.

Data is robust, statistically sound, and controlled. The study design required the expansion of the sections 'material and methods' and 'results', making them extensive, although clear and well written. This led to the manuscript having more than 7,500 words. Since there are no restrictions on the part of the Peer journal, on the number of words, I do not see this as a problem.

The subject itself arouses interest in the scientific community and is far from been fully understood. The authors chose a paradigm to test their hypothesis and remain faithful to it. Considering the way the manuscript was organized, the data presented and discussed, I have nothing to add that can change in an impactful way the message brought by the study.

I take this opportunity to congratulate the authors and, as a reviewer of this important scientific journal, manifest approval and agreement so that the manuscript is approved in its current format.

Minor correction:
Line 27. In the ‘abstract’, please put the name in full and then the abbreviation the first time the term 'conditioned stimulus' appears. The same should be done in Line 162, for the term ‘intertrial interval (ITI)’.
Line 533. “...This means that is our study...”. Replace with ...This means that in our study...

·

Basic reporting

Manuscript is well-written, references provide a clear state of art in the research field. Figures and tables are clear and easy to follow. Abstract needs some improvement, as well as a better basis for the hypothesis of the study.

Experimental design

Research question was clearly defined and evokes for additional research. Some methods information needs to be reviewed or strengthened.

Validity of the findings

Manuscript fails to explore the practical implications of the conclusions.

Additional comments

Abstract

Terms and descriptions are superficial for the first contact with the study. It is not possible for the reader to scale and fully understand the experimental protocols and the determinant variables of analysis of the study. I suggest a more detailed description of the methods, within the allowed word limit.

Introduction
Lines 86-88. Why this hypothesis was established? The first hypothesis is clearly linked to Haddad’s study, but not this one. Could the authors emphasize the basis for this hypothesis as well?

Methods
How sample size was defined? Please, provide information about the participants firstly. I suggest to present table 2 before table 1.
How exclusion criteria were checked? Self-reported? Please, add this information.
It is not clear if all participants performed in all groups or if they were allocated in different groups. Please, emphasize that in the methods section.
Line 133 – “half of the participants” – half of 51? Please, identify in Table 1 the sample size of each group.
Lines 186 – 191 – In the second block, joystick position was on the left or on the right. Therefore, before the trial the participant knew the direction to move the joystick. It is unclear why tone stimulus was necessary.
It is unclear if the pain was randomized inside each block. For example: in the location group, for half of the participants, to move right would always result in pain? Why this protocol was used? One can argue that, in daily activities, people know the movements that result in pain resulting in fear or in avoidance to perform it. Did the authors think about in a ‘control group’, that knew previously the direction that would cause pain?

The practical implications of the study could be strengthened. Authors affirm in the introduction that pain-related fear significantly predicts physical performance and functional disability and that such fear is often more disabling than the pain itself. What conclusions about the differential spatiotopic information can be drawn and eventually used for pain/disability treatments?

Reviewer 3 ·

Basic reporting

This manuscript is generally very well written. The literature is sufficiently informative, as is method and analysis. The tables and figures are excellent and he raw data is provided to allow open re-analysis. The rationale is clear and the hypotheses relate clearly to this and to the conclusions.

Some serious but easily addressed issues in reporting are as follows.

I find Table 1 very confusing. I seem to understand the procedure from the text - but the table does not reflect the text on lines 140-147. Why does the table not show fear conditioning and probes for generalisation for the movement group? Surely the far right column, for example, should list CS presentations for the movement group as these were used as GCs probe stimuli for that group? It is also not clear from the table that startle probes were ALSO part of the fear conditioning phase.

What I understand to be the precise procedure for taking expectancy ratings (inline) is contradicted by Figure 1 suggesting that the text is not sufficiently clear and is not directly related to Figure 1.

In line 209, allusion is made to in-line stimulus ratings but not enough procedural detail is given to allow reader to ascertain how these may have interfered with conditioning trials - this may be critical. It is provided a little later but a flag should be provided earlier for this.

In discussing generalisation, reference is made for the first time to a first and second generalisation test, but there is no mention of this up to this point. I guess it is simply the arbitrary division of the probe trials into two halves - referred to previously as blocks - , and is so likely inconsequential but it would help the reader greatly to not worry that there is some other procedural aspect they have missed.

Line 65 - break up sentence.

Lines 76-78 also have missing words and are too cumbersome.

Line 339 has a crucial word or two missing.

Experimental design

This study has a very elegant and simple design. The mastery of the stimulating and recording equipment is satisfactory and reported well. The study is vert well conducted with no confounds that I can see, but there are compromising procedures. Specifically, in the current inline ratings procedure, if I understand it correctly, you may expect to see a blocking or overshadowing effect whereby the rating Q and response acquire conditioned fear functions that may overshadow the effects of the subsequent CS+. It does not take from the differential effects across conditions and stimuli, but it is not good practice if we can avoid it and it is compromising conditioning to a greater extent, I would argue, than the starting point position of the joystick, which the authors discuss in the Discussion section. This could be acknowledged in the Discussion.

Validity of the findings

Generally the conclusions are solid and relate to the hypotheses. So it passes in that regard. I was surprised, however, not to see more speculation on how these results may relate to our understanding of real world clinical issues around plain management and dysfunction.

One bigger issue is that the question is not as novel as it sounds, but it is sufficiently novel for publication if the authors are clear about the difference between discovering a new effect and discovering a new process. The former is not as exciting as the latter. Specifically, the generalisation shown here, and the effect of overlapping features, is of the same type shown in several previous studies, but is novel (not an essential feature of the MS I know), to the extent that it also involves testing the effect of overlapping features on that acquisition and generalisation. So the results are not telling us much new in terms of conditioning and generalisation processes but they ARE interesting form the point of view of understanding responses to pain. This is not very clear unless the reader is quite expert in the fear conditioning literate and the outcome may come across as over stated. The exciting outcome, is the use of well understood effects, to interpret real life issues relating to movement reduction etc in clinical pain populations. I felt the manuscript would benefit form leaning more in that direction and ensuring that the reader does not mistake a advance in application of principles for an conceptual advance at the process level.

Additional comments

Overall this is an excellent study that is very well reported and deserving of publication.

---

## Round 0.2 · accepted · Accept

Congratulations on your paper accepted for publication.

# ·

Basic reporting

I see no need to re-review this manuscript. In my opinion the manuscript is ready for publication in its current version.

Experimental design

I see no need to re-review this manuscript. In my opinion the manuscript is ready for publication in its current version.

Validity of the findings

I see no need to re-review this manuscript. In my opinion the manuscript is ready for publication in its current version.

Additional comments

I see no need to re-review this manuscript. In my opinion the manuscript is ready for publication in its current version.

·

Basic reporting

I would like to thank the authors for the relevant responses and the revisions submitted.
The new version of the manuscript attends all the questions I raised previously.
Manuscript is well-written, references provide a clear state of art in the research field. Figures and tables are clear and easy to follow.

Experimental design

Research question was clearly defined and evokes for additional research. The additional information provided in the methods section after review allowed to better understand all the protocols designed.

Validity of the findings

No comment.